# Electroceuticals and Magnetoceuticals in Gastroenterology

**DOI:** 10.3390/biom14070760

**Published:** 2024-06-26

**Authors:** Gengqing Song, Roberta Sclocco, Amol Sharma, Ingrid Guerrero-López, Braden Kuo

**Affiliations:** 1Division of Gastroenterology & Hepatology, MetroHealth Medical Center, Case Western Reserve University, Cleveland, OH 44109, USA; songgavin2010@gmail.com; 2Department of Physical Medicine and Rehabilitation, Spaulding Rehabilitation Hospital, Harvard Medical School, Charlestown, MA 02129, USA; rsclocco@mgh.harvard.edu; 3Division of Gastroenterology, Massachusetts General Hospital, Boston, MA 02114, USA; 4Division of Gastroenterology & Hepatology, Department of Medicine, Medical College of Georgia, Augusta University, Augusta, GA 30912, USA; amosharma@augusta.edu; 5Faculty of Medicine, University of Vic-Central University of Catalonia, 08500 Vic, Spain; iguerrero.med@gmail.com

**Keywords:** neuromodulation, vagal nerve stimulation, magnetic neuromodulation, acupuncture, electroacupuncture, gastrointestinal motility disorders, inflammatory bowel disease

## Abstract

In the realm of gastroenterology, the inadequacy of current medical treatments for gastrointestinal (GI) motility disorders and inflammatory bowel disease (IBD), coupled with their potential side effects, necessitates novel therapeutic approaches. Neuromodulation, targeting the nervous system’s control of GI functions, emerges as a promising alternative. This review explores the promising effects of vagal nerve stimulation (VNS), magnetic neuromodulation, and acupuncture in managing these challenging conditions. VNS offers targeted modulation of GI motility and inflammation, presenting a potential solution for patients not fully relieved from traditional medications. Magnetic neuromodulation, through non-invasive means, aims to enhance neurophysiological processes, showing promise in improving GI function and reducing inflammation. Acupuncture and electroacupuncture, grounded in traditional medicine yet validated by modern science, exert comprehensive effects on GI physiology via neuro-immune-endocrine mechanisms, offering relief from motility and inflammatory symptoms. This review highlights the need for further research to refine these interventions, emphasizing their prospective role in advancing patient-specific management strategies for GI motility disorders and IBD, thus paving the way for a new therapeutic paradigm.

## 1. Introduction

Electrical and magnetic neuromodulation is increasingly recognized as a valuable alternative in gastroenterology [1,2,3]. This approach is pivotal for addressing a broad spectrum of gastrointestinal (GI) disorders, particularly those affecting GI motility [2] and inflammatory bowel disease (IBD) [4,5]. The core concept of neuromodulation in gastroenterology is its ability to modify the neural activity of the GI system, offering potential benefits in conditions where traditional treatments may not be fully effective.

This review will explore several key neuromodulation techniques and their applications in gastroenterology. Vagal nerve modulation (VNS), which includes methods such as transcutaneous auricular (taVNS) and cervical vagal nerve stimulation (tcVNS), targets the vagus nerve to regulate brain–gut communication, influencing GI motility, sensation, and inflammatory responses [5]. Additionally, magnetic neuromodulation therapies like pulsed electromagnetic field therapy (PEMF) and repetitive transcranial magnetic stimulation (rTMS) provide non-invasive stimulation of the nervous system, showing potential benefits in enhancing angiogenesis, neurogenesis, and modulating cortical excitability relevant to GI conditions [3,6,7]. Techniques, such as traditional acupuncture, electroacupuncture, and modern non-needle transcutaneous electrical acustimulation (TEA), exert multifaceted effects on neuro-immune-endocrine pathways, impacting motility, inflammation, visceral hypersensitivity, and gut microbiota [5]. This review delves into the devices, parameters, and treatment regimens used in neuromodulation, emphasizing customization based on patient needs and tolerability. It also examines the promising effects of these methods in managing GI motility disorders, IBD, and other related conditions, supported by clinical evidence and study findings.

Through this comprehensive synthesis of current knowledge and research, this study’s aim is to highlight the emerging role of neuromodulation as a valuable complement or alternative to traditional treatments in the field of gastroenterology.

## 2. Vagal Nerve Modulation

Vagal nerve modulation (VNM) emerges as a promising therapeutic strategy for a range of GI disorders. The vagus nerve, a key component of the parasympathetic nervous system, plays a vital role in brain–gut communication, influencing GI functions. Disruptions in this communication can contribute to chronic GI conditions like dysmotility and inflammation. VNM offers a novel approach to address these challenges by restoring balance and functionality to the vagus nerve pathway. This section will explore the mechanisms by which VNM exerts its effects, delve into the technological advancements enabling its application, and critically evaluate its clinical effects and potential based on current research.

### 2.1. Mechanisms of Action of Vagal Nerve Modulation

The vagus nerve plays a crucial role in the regulation of brain–gut interactions, relaying signals between the gut and the brain. Dysregulated gut–brain signaling, often involving disturbances in vagal regulation, is thought to contribute to various gastrointestinal disorders of gut–brain interactions (DGBI), including functional dyspepsia, gastroparesis, or irritable bowel syndrome. The mechanisms of action of vagal nerve modulation for gastrointestinal disorders involve the stimulation of both the afferent and efferent fibers of the vagus nerve, which is a key component of the parasympathetic nervous system. The afferent fibers carry information from the gastrointestinal tract to the brainstem, while the efferent fibers transmit information from the brainstem to the gastrointestinal organs, therefore modulating motility and sensation. Transcutaneous electrical stimulation techniques, such as taVNS, TEA (see Section 4.), percutaneous electrical nerve field stimulation (PENFS), and tcVNS, work by modulating the autonomic nervous system [1,8,9,10] (Figure 1). Also, stimulation of the interstitial cells of Cajal, which are essential for generating and propagating electrical slow waves and coordinating gut contractions, is another suggested mechanism [11]. These modalities have also been shown to activate cholinergic anti-inflammatory pathways, reducing pro-inflammatory cytokine production [12].

### 2.2. Devices, Parameters, and Treatment Regimens in Use

The vagus nerve has a major influence on gastric motility, and gastrointestinal emptying is regulated by both the vagal and splanchnic pathways, as well as the intrinsic innervation via the enteric nervous system. Vagal nerve stimulation for gastrointestinal disorders is typically performed through various non-invasive transcutaneous or percutaneous electrical stimulation techniques. Some examples include the following:

Transcutaneous Auricular Vagal Nerve Stimulation (taVNS): Small surface electrodes are placed near the cymba concha in the external auricle to elicit a diffuse stimulation field, activating the auricular branch of the vagus nerve (ABVN), which is predominantly afferent. A therapeutic stimulation frequency of between 0.5 and 30 Hz is typically used, with 25 Hz being the most commonly used frequency to enhance vagal efference and increase motility [13,14]. The main target of taVNS is the nucleus tractus solitarius in the medulla region of the brainstem, and recent studies combining taVNS with functional imaging reported increased target engagement when delivering the stimulation at 100 Hz [15]. The stimulation intensity is usually adjusted based on patient tolerability. 

Percutaneous electrical nerve field stimulation (PENFS): Similar to taVNS, PENFS is also delivered in the external auricle, and it targets central pathways through branches of the vagus nerve and three other cranial nerves (V, VII, and IX) that innervate the external ear. A commercial device (IB-STIM, NeurAxis, Carmel, IN, USA) is FDA-cleared for adolescent patients with functional abdominal pain associated with irritable bowel syndrome. Electrical stimulation is delivered percutaneously using sterile, titanium needles, at 3.2 V, with a rectangular pulse wave and alternating frequencies (1 ms pulse of 1 and 10 Hz) every 2 s. The battery lasts 5 days, during which the device is programmed to deliver stimulation in cycles of 2 h on and 2 h off for 120 h.

Transcutaneous Cervical Vagal Nerve Stimulation (tcVNS): This is performed non-invasively using a portable powered stimulator with steel contact electrodes, applied to the neck, delivering electrical stimulation to the cervical branch of the vagus nerve (mix of afferent and efferent fibers). As an example, ElectroCore has developed a cervical transcutaneous stimulator called gammaCore, which utilizes two stainless-steel round discs as skin contact surfaces to deliver a low-voltage electrical signal (5 kHz sine waves each lasting 200 ms, repeated once every 40 ms or 25 Hz) to the cervical vagus nerve on the neck. The device can provide a programmable number of stimulation cycles, each lasting 120 s, and the stimulation intensity can be increased until a mild but stable contraction of the ipsilateral orbicularis oris muscle is obtained. The treatment regimen for these techniques varies based on the specific patient’s need and tolerability, but, frequently, they are delivered in less than 60 min each time and from once every other day to twice daily. The duration for these interventions also varies and can be as long as several weeks, based on individual response.

### 2.3. Impact on Major GI Diseases

Motility: Vagal nerve modulation can have a positive impact on a variety of gastrointestinal motility disorders where pharmacological therapies do not offer adequate symptom relief. For instance, short-term tcVNS led to improved cardinal symptoms and accelerated gastric emptying in a subset of patients with idiopathic gastroparesis [16]. taVNS has been reported to prevent and reverse esophageal pain hypersensitivity [14], as well as reduce the symptoms of functional dyspepsia including improvements in gastric accommodation and reductions in gastric dysrhythmia [13]. In both cases, the effects are mediated by an increase in vagal efferent activity. These studies suggest that vagal nerve modulation can provide an alternative treatment for patients suffering from these gastrointestinal disorders. However, more research is needed to further investigate and solidify these findings.

Inflammation: Vagal nerve modulation is believed to have therapeutic potential in the management of gastrointestinal disorders such as inflammatory bowel diseases, which include Crohn’s disease and ulcerative colitis. This is due to the vagus nerve anti-inflammatory properties through its afferents and efferents, including the cholinergic anti-inflammatory pathway. Vagus nerve stimulation could represent a non-drug therapeutic approach as an alternative to conventional anti-TNF-alpha drugs [17]. It has the advantage of being devoid of major side effects and is less expensive than biologics. tVNS has been reported to reduce paracellular permeability of the small intestine induced by stress peptide corticotropin-releasing hormone [18]. In pilot studies, invasive VNS delivered via an implanted electrode wrapped around the left cervical vagus nerve has shown promising results, reducing the inflammatory symptoms of these diseases and helping to repair the bowel mucosa [19]. However, more research is needed to further establish parameters for optimal efficacy and safety.

### 2.4. Review of Key Clinical Studies

Several pilot studies have explored the use of vagus nerve stimulation in patients with treatment-resistant gastrointestinal disorders, including gastroparesis and IBD. In the case of gastroparesis, tcVNS has shown promise in improving symptoms. Two pilot studies, one conducted in the UK and another in the USA, reported significant improvements in symptoms, such as bloating, abdominal pain, and gastric emptying time, after a few weeks of tcVNS. The response rate in these studies ranged from 43% to 67% [16,20]. A pilot study conducted in France focused specifically on Crohn’s disease patients. The authors found that invasive VNS resulted in improved clinical remission rates and reduced disease activity scores in these patients [19,21]. Overall, these studies support the potential effectiveness of vagus nerve stimulation in treating gastrointestinal disorders and associated pain.

PENFS has recently been proposed as a therapeutic option to improve comorbidities in children with cyclic vomiting syndrome. The treatment was shown to improve anxiety, sleep, and several aspects of quality of life, with long-term benefits for anxiety [22].

A randomized, sham-controlled trial evaluated the effects of a 4-week taVNS treatment on constipation-predominant irritable bowel syndrome. Compared to sham, taVNS significantly increased complete spontaneous bowel movements per week and decreased symptoms associated with irritable bowel syndrome, including abdominal pain [23]. Another study explored the acute and longitudinal (2 weeks) effects of taVNS in patients with functional dyspepsia. The authors found that taVNS acutely improved gastric accommodation, increased the percentage of normal gastric slow waves, and enhanced vagal activity compared to sham. Longitudinal effects also included a reduction in symptoms of dyspepsia, as well as in anxiety and depression [13].

## 3. Magnetic Neuromodulation

Magnetic stimulation provides a safe, non-invasive neuromodulation alternative that does not require direct contact with the target nerve via a connecting lead (Figure 2). This technique includes two main types: PEMF therapy and rTMS [3]. Both modalities provide distinct advantages for treating neurological [6], musculoskeletal [24], and GI conditions. This section will discuss the novel application of these magnetic therapies, particularly focusing on peripheral repetitive magnetic stimulation for managing fecal incontinence (FI) and diabetic gastroparesis (DGp), exploring the clinical implications and underlying mechanisms.

### 3.1. Mechanisms of Action of Magnetic Stimulation

The exact mechanism of neuromodulation in response to magnetic stimulation remains unknown. Observations from exploratory studies from both PEMF and rTMS can be extrapolated to suggest the mechanism of action. PEMF exposure significantly improved angiogenesis in both normal and diabetic mice, driven by the five-fold increase in fibroblast growth factor ß-2 (FGF-2) by endothelial cells [25,26]. Extremely low-frequency PEMF promotes neuronal differentiation, neurite outgrowth, synaptic plasticity, long-term potentiation, and neurogenesis in the hippocampus of mice by upregulating Ca_v_1-channel activity, even in ischemic states [27,28,29,30]. Furthermore, extremely low-frequency PEMF-induced neurogenesis increases the proportion of inhibitory GABAergic neurons [31]. rTMS can influence cortical excitability; however, studies have been mixed on its effects. In healthy subjects, rTMS resulted in homeostasis of cortical inhibition, with decreased cortical inhibition in subjects with greater baseline inhibition and increased cortical inhibition with lower baseline inhibition [32].

Neuromodulation of the innervation of the GI tract can occur through three pathways—vagal, spinal, and sacral [33]. The vagal innervation extends from the distal esophagus to the descending colon and is predominantly parasympathetic. Thoracolumbar spinal nerves coursing the sympathetic chain ganglia provide the main sympathetic innervation to the foregut, midgut, and most of the hindgut. Some axons of cervical sympathetic postganglionic neurons are also carried by the vagus nerve. The sacral nerves innervate the distal colorectum. Vagal afferent neurons are unmyelinated, slow-conducting C-fibers with dense plexi of mucosal free endings at the gut epithelium and receive hormonal signals from the enteroendocrine cells (CCK, 5-HT, ghrelin, GLP-1). Vagal efferents originate from the nucleus ambiguous and dorsal motor nucleus of the vagus and regulate peristalsis and secretion of the GI tract. Thoracolumbar spinal afferents carry signals related to pain and distension from the GI tract to the somatosensory cortex through a three-neuron chain [34]. Spinal afferents are the first-order neurons with cell bodies in the dorsal root ganglion (DRG) and synapses in the dorsal horn. Ascending visceral afferents, second-order neurons, send projections to the brain through gracile fasciculus of the dorsal column medial lemniscus (DCML) pathway in the spinal cord [35]. Thoracolumbar spinal sympathetic efferents inhibit GI transit, inducing contraction in sphincteric regions, and inhibiting vasoconstrictor and secretomotor pathways [36]. The inhibition of sympathetic spinal nerves can also be anti-inflammatory and involved in appetite regulation [37,38]. Visceral hypersensitivity and the three-neuron chain are the ‘holy grail’ in understanding DBGI [39]. Sacral nerves carry both parasympathetic and sympathetic innervation to the hindgut as well as somatic efferents, regulating motility and blood flow to the colorectum.

### 3.2. Devices, Parameters, and Treatment Regimens in Use

Magnetic neuromodulation devices, notably in the forms of PEMF and rTMS, allow for a broad range of parameter adjustments to cater to specific therapeutic needs.

PEMF Devices: These devices involve placing the patient in a pulsing electromagnetic field, similar to the setting of an open MRI [3]. This field can be adjusted to various frequencies, ranging from ultra low (<3 Hz) to very low (30 kHz–300 kHz), depending on the therapeutic goals. The exposure time can vary from a few minutes to several hours, allowing for flexible treatment schedules. Traditionally used in orthopedics for bone healing [24], PEMF is being explored in gastroenterology for its effects on tissue repair and inflammation modulation.

rTMS Devices: These devices utilize large electromagnetic wire coils connected to a generator. These coils are placed on the scalp in precise locations to target specific brain regions. The rTMS device delivers hundreds of rapidly alternating magnetically inducted electronic fields, affecting neurons and cortical excitability. The stimulation frequency can be adjusted to induce cortical inhibition at low frequencies (≤1 Hz) or excitation at high frequencies (>1 Hz). rTMS has strong Level A evidence supporting its use in conditions like neuropathic pain, depression, and motor recovery in the post-acute stage of stroke [6].

Both PEMF and rTMS are characterized by their non-invasive nature and the ability to tailor treatment parameters to individual needs, enhancing both efficacy and patient tolerability. These devices allow for precise control over treatment variables, which is critical for achieving optimal therapeutic outcomes in complex GI conditions.

### 3.3. Impact on Major GI Diseases

Magnetic neuromodulation, particularly through techniques like rTMS and targeted spinal magnetic therapies, has shown promising results in the management of complex GI disorders such as FI [40] and DGp [41]. Unlike traditional methods such as VNS and gastric electrical stimulators, which are limited by patient tolerance, cost, and invasiveness, newer approaches like transcutaneous VNS (tVNS) offer less-invasive alternatives. However, optimal tVNS settings often utilize moderate to high intensities to generate increased amplitudes of vagal sensory evoked potentials (VSEPs) [42], although many protocols employ lower intensities (1-3 mA) tailored to the patient’s pain threshold [43].

The advantage of repetitive magnetic stimulation is non-invasive, does not require the implantation of leads or battery generators, and is painless, especially at the higher intensities needed for effective neuromodulation [44]. Magnetic stimulation can deliver a greater intensity during treatment compared to traditional electrical stimulation and allows for the modulation of deep brain structures, though challenges remain in targeting structures deeper than 3 cm from the surface, despite advancements in stimulation protocols and coil technology [45]. The efficacy of rTMS is also influenced by individual anatomical factors such as gyral folding patterns and white matter anisotropy, which affect the distribution and effectiveness of the induced electrical fields [46].

Fecal Incontinence (FI): FI is a prevalent condition where patients experience the recurrent, unintended passage of mucus and/or liquid or solid stool that affects 15% of the Western population and significantly impacts quality of life [47]. Current treatment options for FI include labor-intensive home or in-office biofeedback therapy [48], dextronomer injection, or sacral nerve stimulation (SNS), requiring surgical implantation, and remain unsatisfactory for patients [49]. Neuropathy in FI is confined to the lumbosacral innervation of the anorectum, not involving corticospinal tracts of the bidirectional gut–brain axis [50]. Traditional treatments for FI often require invasive procedures or have limited effectiveness. Emerging therapeutic strategies like translumbosacral neuromodulation therapy (TNT) utilize repetitive magnetic stimulation aimed at the lumbosacral nerves, which are crucial for anorectal function. By non-invasively stimulating these nerves, TNT has demonstrated potential in improving the neuromuscular control of the anorectum, significantly reducing the symptoms of FI and improving patients’ quality of life, without the risks associated with surgical interventions [40].

Diabetic Gastroparesis (DGp)*:* DGp is a debilitating condition, with patients suffering from nausea, vomiting, early satiety, postprandial fullness, and abdominal pain associated with increasing hospitalizations and impaired quality of life [51]. Diagnosis requires the absence of mechanical obstruction and evidence of delayed gastric emptying. Metoclopromide is the only FDA-approved medication for gastroparesis, limited in its use due to a black box warning of tardive dyskinesia, and gastric electrical stimulation is only approved for compassionate use. Sixteen pharmacological drugs have failed to show improvement over placebo in DGp subjects [52]. Treatment options for DGp are lacking. Despite expert management, only 28% of patients had a significant reduction in their symptom scores [53]. Thoracic Spinal nerve Neuromodulation Therapy (ThorS-MagNT) leverages repetitive magnetic stimulation targeting thoracic spinal nerves to modulate the underlying pathological mechanisms of DGp (Figure 2). Initial clinical studies suggest that ThorS-MagNT can effectively reduce symptoms, such as nausea, vomiting, and abdominal pain, by enhancing gastric motility and potentially modifying the gastric neural reflexes involved in gastroparesis.

### 3.4. Review of Key Clinical Studies

The integration of magnetic neuromodulation into gastroenterology has marked a significant advancement in the treatment of complex disorders like FI and DGp. This section delves into pivotal clinical studies that explore the efficacy and mechanisms of TNT and ThorS-MagNT, emphasizing their potential to transform current treatment approaches.

TNT is a repetitive magnetic stimulation treatment, shown to reverse spinal neuropathy of the lumbosacral nerves, that was developed for the treatment of FI [7]. Using single pulse stimulations over the lumbosacral nerve roots, translumbosacral anorectal magnetic stimulation (TAMS) determines the degree of lumbosacral spinal neuropathy and threshold intensity used for repetitive magnetic stimulation prior to TNT [54]. In a randomized dose-ranging study, 1 Hz TNT delivering 600 repetitive magnetic stimulations at each of the four lumbosacral sites weekly for six weeks was found to be superior to 5 or 15 Hz [40]. A multi-site randomized sham-controlled study is ongoing to determine safety and short-term and long-term efficacy in FI subjects (NCT03899181).

ThorS-MagNT is a promising repetitive magnetic stimulation treatment targeting bilateral spinal nerves at T7 for subjects with DGp [41]. In a proof-of-concept study, a novel, neuromodulation treatment, Thoracic Spinal Nerve Magnetic Neuromodulation Therapy (ThorS-MagNT), was examined in a small, uncontrolled trial of adults with moderate–severe DGp [41]. ThorS-MagNT is administered by delivering 1200 magnetic stimulations to each side daily for 5 consecutive days. It showed a 75% reduction in the total score of the American Neurogastroenterology and Motility Society Gastroparesis Cardinal Symptom Index - Daily Diary (ANMS GCSI-DD), which is a 7-day composite measure of the main symptoms of gastroparesis. This response improved to >95% 2 weeks after treatment, demonstrating a sustained response and suggesting neuroplasticity of spinal nerves. A multi-site randomized sham-controlled phase Ib study is ongoing to further determine safety and efficacy (NCT05273788). In this study, optimal treatment demonstrated a 75–95% improvement in total GCSI that persisted > 21 days post-treatment in DGp subjects.

In summary, targeting spinal nerves with repetitive magnetic stimulation is promising, well-tolerated, and non-invasive for the treatment of conditions associated with perturbations in the gut–brain axis. Preliminary evidence is reassuring for repetitive magnetic treatments to improve the conditions of FI and DGp in patients, guided by supporting mechanistic evidence and a strong rationale.

## 4. Acupuncture, Electroacupuncture, and Transcutaneous Electrical Acustimulation (TEA)

Acupuncture, electroacupuncture, and TEA represent pivotal modalities in the evolving landscape of GI therapy. Grounded in both traditional practices and modern clinical research, these treatments offer profound insights into the neuro-immune-endocrine interactions that modulate many GI disorders. This section elucidates the mechanisms of action behind these therapies, detailing how they influence GI motility, inflammation, visceral hypersensitivity, barrier function, and microbiota composition. By exploring the scientific basis and clinical applications of acupuncture and electroacupuncture, we can better understand their role in offering non-invasive, effective alternatives to traditional GI treatments, thereby opening new avenues for patient care and symptom management.

### 4.1. Mechanism of Action of Acupuncture, Electroacupuncture, and TEA

Acupuncture and electroacupuncture exert multifaceted pathophysiological effects on the gastrointestinal (GI) system through complex neuro-immune-endocrine mechanisms, including GI motility [55,56], inflammation [57,58,59], visceral hypersensitivity [60,61,62], GI barrier function [63,64,65], and microbiota [66,67,68,69].

A primary mechanism is the enhancement of GI motility by stimulating specific acupoints such as PC6 (Nei Guan) and ST36 (Zu Sanli). These points interact with the autonomic nervous system [70], particularly the vagal pathways, leading to improved esophageal motility [71], LES pressure [72,73,74], gastric emptying [75,76,77,78,79,80,81,82], small intestinal motility (contractions and transit) [83,84], and colonic motility [85,86] in both animals and humans [79,80].

In terms of modulating the inflammatory response, which is central to many GI disorders, acupuncture techniques have shown the capability to balance the immune system. This includes reducing the activity of inflammatory enzymes like neutrophil myeloperoxidase, suppressing pro-inflammatory cytokines (TNF-α, IL-1β) [87,88], and inducing anti-inflammatory cytokines like IL-10 [87,89]. This modulation is crucial for managing inflammation-driven GI conditions such as IBD [90,91].

Acupuncture also addresses visceral hypersensitivity, commonly seen in GI motility disorders, by modifying both afferent and efferent neural pathways [92,93,94,95]. This involves attenuating afferent nerve sensitization via the transient receptor potential vanilloid type 1 (TRPV1) and dorsal root ganglion (DRG) neurons [96,97] and enhancing vagal efferent activity [98], leading to an anti-inflammatory state and reduced sympathetic nervous system output, thereby mitigating pain and discomfort [98,99].

Furthermore, these therapies play a significant role in maintaining and restoring gut barrier integrity, particularly relevant in IBD [63,64,65,91,100]. This is achieved through the upregulation of tight junction proteins (ZO-1 [63,100], occluding and claudin-1 [100]), reducing intestinal permeability [64].

Crucially, acupuncture has a significant impact on the gut microbiota, an emerging focus in GI health. Treatment leads to shifts in key bacterial populations (increase in Bifidobacterium and Lactobacillus [66,67,69] but decrease in E coli and B fragilis [66]), which in turn influence the gut’s inflammatory environment. Such changes in microbiota composition have been linked to improvements in GI symptoms and inflammatory profiles. This suggests a symbiotic relationship between the gut microbiota and the host’s immune responses, highlighting another layer of complexity in acupuncture’s therapeutic mechanisms.

In conclusion, acupuncture and electroacupuncture’s effectiveness in GI disorders stems from their ability to modulate GI motility, immune response, visceral sensitivity, gut barrier function, and gut microbiota, acting through an intricate network of neuro-immune-endocrine interactions.

### 4.2. Devices, Parameters, and Treatment Regimen in Use

In traditional acupuncture, practitioners insert a fine metallic needle at specific acupoints (Figure 3A) and manipulate it using techniques like thrusting and twisting. Electroacupuncture, however, modernizes this approach by attaching a pulse generator to the needle, providing consistent electrical stimulation (Figure 3B). This method is gaining traction in research due to its uniformity and reproducibility, distinguishing it from traditional manual methods [101,102].

A novel development in this arena is TEA, shown in Figure 3C. This technique uses surface electrodes for electrical stimulation at acupoints, bypassing the need for needle penetration. Research indicates that TEA is as efficacious as traditional acupuncture in mitigating gastrointestinal discomfort and dysmotility issues [103]. Its non-invasive nature allows for ease of use in various environments, including homes and workplaces, enhancing patient acceptability.

In acupuncture, electroacupuncture, and TEA, the stimulation parameters are generally intermittent and can be adjusted according to the specific treatment needs. These parameters typically include a frequency range of 5 to 100 Hertz (Hz), a pulse width of at least 0.1 milliseconds, and an amplitude ranging from 1 milliampere (mA) to 10 milliamperes, contingent upon the patient’s tolerance. This flexibility in the adjustment of parameters allows for a tailored approach to meet individual patient requirements and enhances the effectiveness of the treatment [55,56,57,58,59,60,61,62].

### 4.3. Impact on Major GI Diseases

Acupuncture techniques have shown notable effects in the management of various GI disorders, supported by their neuro-immune-endocrine mechanisms of action.

Motility: These therapies are particularly effective in conditions characterized by altered GI motility, including gastroesophageal reflux disease (GERD), gastroparesis, functional dyspepsia (FD) [1,56]. By stimulating specific acupoints, these treatments enhance motility across the GI tract. For GERD, combining EA at acupoints ST36 and PC6 with proton pump inhibitors (PPIs) significantly improved symptoms of heartburn and acid regurgitation, outperforming doubled doses of the inhibitors alone [56]. In FD, a 4-week EA treatment resulted in sustained symptom relief, with a significant number of patients showing improvements 16 weeks post-treatment [56]. Moreover, in chronic functional constipation, EA substantially increased weekly complete spontaneous bowel movements in a large-scale study, demonstrating its efficacy over placebo [56]. TES has also shown promising effects in enhancing gastric emptying and improving symptoms in gastroparesis patients [1]. Additionally, in post-operative settings, acupuncture has been shown to alleviate postoperative ileus and accelerate bowel recovery following colorectal surgery, further indicating its role in normalizing motility post-intervention [56].

Inflammation: Acupuncture’s role in modulating the immune response is critical in the management of GI inflammation such as IBD [4,5]. Research indicates that acupuncture significantly reduces the serum levels of TNF-α by approximately 30-40%, a key pro-inflammatory cytokine implicated in the pathogenesis of these diseases. Concurrently, it enhances IL-10 levels, a critical anti-inflammatory cytokine, by up to 50%, contributing to a reduction in clinical symptoms and inflammatory activity. These biochemical changes are crucial for both alleviating current flare-ups and maintaining long-term remission. Furthermore, studies have shown that acupuncture improves intestinal barrier function, evidenced by the increased expression of tight junction proteins such as ZO-1 and claudin-1, which are essential for preventing pathogen invasion and subsequent inflammation. These findings are supported by a decrease in clinical relapse rates and an improvement in patient-reported outcome measures in controlled trials.

Visceral Hypersensitivity and Related Conditions: Acupuncture effectively addresses visceral hypersensitivity, a hallmark of conditions like IBS and functional dyspepsia [2,55,56]. This treatment modulates neural pathways by attenuating afferent sensitization and enhancing vagal efferent activity, directly impacting the gastrointestinal tract’s sensitivity to pain and discomfort. Studies have shown that acupuncture can reduce abdominal pain scores by up to 30-50% in patients with IBS, with improvements in discomfort persisting for up to three months post-treatment. Additionally, through the regulation of vagal tone, acupuncture has been found to increase pain thresholds by approximately 40% in patients with functional dyspepsia. The therapy’s impact on the sympathetic nervous system also contributes significantly, with clinical trials reporting a marked decrease in hypersensitivity symptoms, correlating with reductions in sympathetic overactivity. These changes are not only symptomatic but also reflect physiological improvements in gut motility and a decrease in the inflammatory markers often associated with these conditions.

Gut Microbiota Modulation: The impact of acupuncture on gut microbiota composition has been explored in various GI conditions [104,105]. Acupuncture is shown to induce favorable shifts in gut microbiota that are directly linked to improvements in GI motility and inflammatory profiles, which are critical for conditions influenced by dysbiosis. For example, experimental studies have demonstrated that electroacupuncture treatments can significantly increase the populations of beneficial bacteria such as Lactobacillus and Bifidobacterium by over 40%, while reducing harmful species like Clostridium by up to 25%. These microbiotic adjustments correlate with a decrease in inflammatory cytokines and an enhancement in bowel function, evidenced by improved transit times and reduced symptoms of discomfort and bloating. Moreover, a clinical trial involving patients with irritable bowel syndrome (IBS) noted a restoration of the microbiota balance after a series of acupuncture sessions, which was associated with a 30% reduction in clinical symptoms including abdominal pain and irregular bowel movements. These findings underscore the potential of acupuncture and electroacupuncture to modulate gut microbiota, contributing to their therapeutic effects across a spectrum of GI disorders marked by dysbiosis.

In summary, acupuncture and electroacupuncture present clinically significant benefits in managing a range of GI disorders. These advantages manifest as enhanced motility, diminished inflammation, alleviated hypersensitivity, and modulated gut microbiota, supporting the integration of these therapies into conventional treatment plans for a more comprehensive and effective management of GI diseases.

### 4.4. Review of Key Clinical Studies

Regarding GI motility disorders and IBD, a series of clinical trials have highlighted the beneficial effects of acupuncture techniques. These studies encompass a wide array of conditions, including GERD, FD, constipation, ulcerative colitis (UC), and Crohn’s disease, demonstrating the versatile therapeutic potential of acupuncture in managing diverse GI disorders.

Up to 30% of GERD patients do not achieve symptom relief with standard PPI therapy. In this context, acupuncture techniques offer promising results. A study by Meng et al. demonstrated that TEA significantly ameliorated GERD symptoms and enhanced lower esophageal sphincter (LES) pressure by 52.2% compared to sham stimulation [73]. Furthermore, combining TEA with Deep Breathing Training (DBT) has been shown to be more effective in reducing reflux symptom scores than sham combinations, suggesting a synergistic effect that could redefine non-pharmacological treatment strategies [72]. Additionally, meta-analytical data from multiple studies have confirmed that acupuncture improves global GERD symptom scores and quality of life, with notable reductions in symptom severity and frequency [106]. These clinical outcomes not only underline the symptomatic relief provided by acupuncture but also highlight its potential impact on key physiological mechanisms such as reducing esophageal acid exposure and strengthening LES function.

FD is a GI disorder that often proves resistant to conventional treatment strategies. However, recent studies, including a systematic review and meta-analysis by Kwon et al., which analyzed 22 randomized controlled trials, have shown promising results for acupuncture and TEA. This meta-analysis review found that the addition of acupuncture to conventional treatments significantly improved both individual and total symptom scores for FD. Notably, patients receiving acupuncture showed an improvement in the Nepean Dyspepsia Index (NDI), with a mean difference of -7.92 points, suggesting significant symptom relief compared to conventional treatment alone [107]. In FD patients, Liu et al. [108] conducted a double-blind crossover study, revealing that acute TEA increased vagal activity significantly without altering gastric slow waves. With two-week TEA treatment, there was a notable decrease in dyspepsia symptoms and further increase in vagal activity, along with an elevation in serum neuropeptide Y (NPY) levels. Supporting these findings, Ji et al. reported that FD patients treated with TEA experienced significant improvements in dyspeptic symptoms, quality of life, and gastric motility functions [81]. This was further substantiated by Ma et al., who found that synchronized TEA (STEA) was more effective than TEA alone, emphasizing the role of enhanced parasympathetic activity in managing dyspepsia. The total effective rate (TER) in these studies often exceeded 1.29 (95% CI 1.23–1.34), indicating a 29% better outcome in response to acupuncture compared to control groups [109].

Acupuncture techniques have gained attention for their potential in treating constipation, particularly in populations where traditional medications lead to undesirable side effects or insufficient efficacy. A comprehensive meta-analysis by Liu et al. [110] examined thirteen randomized controlled trials encompassing 1437 participants to assess the impact of EA on secondary constipation (that occurs after certain diseases or medications, such as acute stroke or opioids). The findings demonstrate that EA notably improves overall response rates in patients, with a relative risk (RR) of 1.31, indicating a 31% improvement compared to control groups. Additionally, EA was effective in reducing the defecation straining score, with a mean difference (MD) of -0.46, and significantly increased both weekly complete spontaneous bowel movements (CSBMs) and spontaneous bowel movements (SBMs), with MDs of 0.41 and 0.80, respectively. The analysis also addressed EA’s impact on stool consistency and patient quality of life (PAC-QOL scores) but found no significant differences in these parameters compared to control treatments. Importantly, the use of EA did not lead to an increase in adverse events, suggesting a favorable safety profile. However, despite these positive outcomes, the study highlighted the need for higher-quality research to further validate these findings due to existing limitations in study designs and inconsistencies in measured outcomes across different studies. This systematic review underscores the potential of electroacupuncture as a viable treatment alternative for secondary constipation, promoting gastrointestinal motility and improving patient outcomes, without increasing the risk of adverse effects.

UC is a chronic inflammatory bowel disease characterized by periods of exacerbation and remission, presenting significant challenges to conventional therapeutic approaches. A systematic review and meta-analysis evaluating the efficacy of acupoint application in UC treatment provided substantial insights [111]. Analyzing data from 13 randomized controlled trials involving 878 participants, the review demonstrated that acupoint application significantly improved the clinical comprehensive effective rates (risk ratio [RR] 1.13, 95% CI 1.06–1.20) and syndrome effective rates (RR 1.13, 95% CI 1.03–1.24) compared to conventional Western medicine alone. Additionally, the treatment was associated with positive modulations in inflammatory markers, with increases in interleukin-4 (mean difference [MD] 2.62, 95% CI 1.96–3.28) and decreases in interferon-γ (MD −5.38, 95% CI −6.81 to −3.94), suggesting an immunomodulatory effect. Despite these improvements, no significant differences were observed in colonoscopy outcomes or the rate of adverse reactions compared to control treatments, indicating a comparable safety profile. The quality of evidence for these findings ranged from moderate to very low due to inconsistencies and methodological limitations in the included studies, highlighting the need for more rigorous research to conclusively determine the role of acupoint application in UC management.

For Crohn’s disease, fewer trials are available, yet they provide insightful results. A German study [112] revealed that acupuncture led to a significant reduction in Crohn’s Disease Activity Index (CDAI) scores and improvements in general well-being compared to minimal acupuncture. Another randomized controlled trial [113] showed that acupuncture combined with moxibustion was more effective in reducing CDAI scores and enhancing quality of life, with a substantial proportion of patients achieving remission.

In summary, these meta-analysis and clinical trials collectively indicate the potential of acupuncture as effective treatments for various GI disorders. These therapies have shown improvements in clinical symptoms, reductions in inflammation, and the modulation of autonomic activity across a range of GI conditions. However, the variability in study designs and quality highlights the need for more standardized and robust research to further establish their role in GI treatment regimens.

## 5. Disadvantages and Drawbacks of Electroceuticals and Magnetoceuticals

Although electroceuticals and magnetoceuticals offer promising treatment options for GI disorders, several limitations and potential drawbacks should be considered. One significant limitation is that most of the clinical studies on vagal nerve electrical and magnetic neuromodulation involve small sample sizes and are often conducted as single-center studies. This can limit the generalizability and robustness of the findings. Another important limitation is the variability in patient response to these treatments. Individual differences in anatomy, disease severity, and concurrent medications can influence the efficacy of these neuromodulation techniques, leading to inconsistent outcomes among patients. Additionally, the precision required for device placement and parameter settings can pose challenges, as slight deviations may result in suboptimal therapeutic effects or adverse outcomes.

### 5.1. Stimulation Parameters

Stimulation parameters, including frequency, intensity, and duration, are critical for the success of neuromodulation techniques. Although there were recommended parameters in small trials from each single center, there is currently no consensus on the optimal parameters, necessitating customization for each patient. This variability can significantly impact treatment outcomes, complicating efforts to standardize neuromodulation therapies.

### 5.2. Safety Concerns

Although the safety profile of electroceuticals and magnetoceuticals is generally favorable, certain risks and adverse events may be associated with their use. For electroceuticals, potential side effects include skin irritation or burns at the electrode sites, discomfort during stimulation, and, in rare instances, interference with implanted medical devices such as pacemakers. Magnetic neuromodulation techniques may cause transient headaches, dizziness, or scalp discomfort. Additionally, there is a theoretical risk of inducing seizures, particularly in patients with a history of epilepsy or other neurological conditions.

### 5.3. Drawbacks

Moreover, the requirement for specialized equipment and trained personnel to administer these therapies can limit accessibility and increase the cost of treatment.

## 6. Conclusion

The exploration of neuromodulation techniques in gastroenterology reveals significant advancements and potential in managing GI disorders. VNS, including taVNS and tcVNS, emerges as a key player in modulating brain–gut communication, influencing GI motility, sensation, and inflammation. Magnetic neuromodulation, encompassing PEMF and rTMS, offers a non-invasive, innovative approach, showing promise in enhancing neurophysiological processes relevant to GI health. Acupuncture and electroacupuncture, with their roots in traditional medicine, demonstrate multifaceted effects on the GI system, impacting motility, inflammation, visceral hypersensitivity, and microbiota.

Each neuromodulation method presents unique strengths and applications. VNS stands out for its direct impact on the parasympathetic nervous system, offering targeted intervention for GI motility and inflammatory conditions. Magnetic neuromodulation, while non-invasive, provides broader systemic effects, potentially beneficial in complex, multifactorial GI disorders. Acupuncture and electroacupuncture offer versatility and holistic treatment approaches, addressing both physiological and psychosomatic aspects of GI conditions.

The future of neuromodulation in gastroenterology is promising, with potential advancements and broader applications. Continued research and clinical trials are essential to deepen our understanding of these techniques, optimize treatment parameters, and validate long-term efficacy. Personalized medicine approaches, integrating patient-specific factors and responses to neuromodulation, could enhance treatment effectiveness and patient satisfaction. Additionally, the integration of neuromodulation with conventional treatments may offer synergistic effects, leading to more comprehensive management strategies for GI disorders.

In conclusion, neuromodulation techniques represent a frontier in gastroenterological treatments, with each method offering unique benefits. Future research and clinical applications are poised to further transform the management of GI disorders, potentially improving outcomes for patients with conditions that are currently challenging to treat effectively.

## Figures and Tables

**Figure 1 biomolecules-14-00760-f001:**
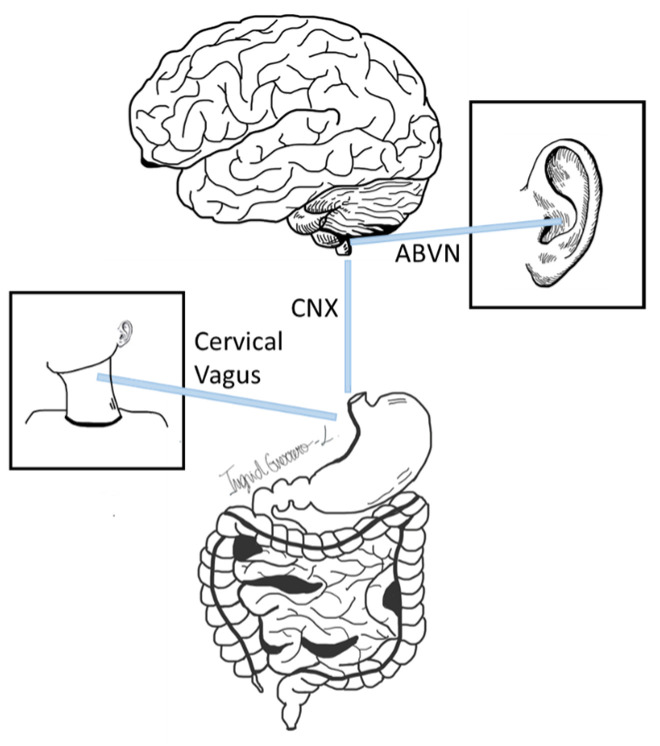
Schematic illustration of peripheral targets for non-invasive vagus nerve stimulation. CNX: Cranial Nerve X (Vagus Nerve); ABVN: auricular branch of the vagus nerve.

**Figure 2 biomolecules-14-00760-f002:**
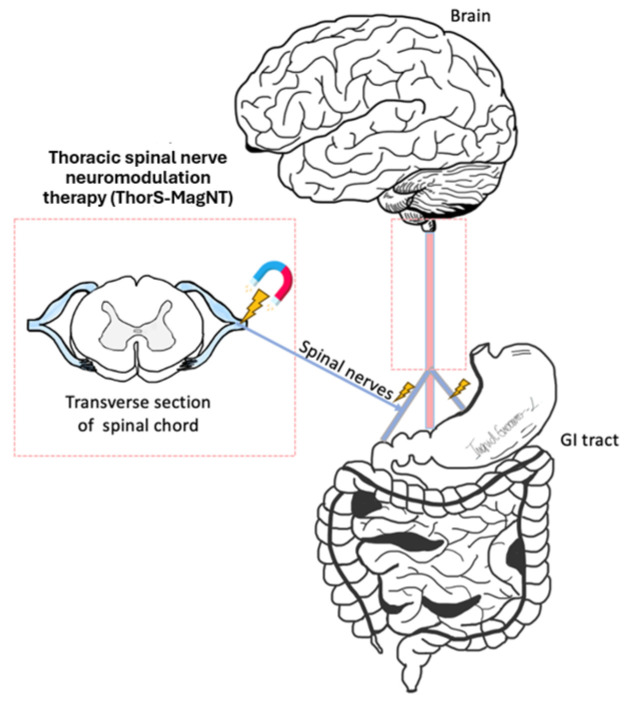
Schematic illustration of peripheral targets for non-invasive magnetic stimulation.

**Figure 3 biomolecules-14-00760-f003:**
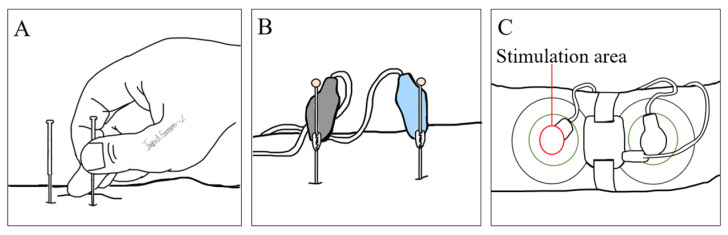
Schematic illustration of acupunture techniques. (**A**) Traditional acupuncture with hand-manipulated needles; (**B**) electroacupuncture using electrical stimulation of needles; (**C**) transcutaneous electrical acustimulation (TEA) via skin electrodes, without needles.

## Data Availability

No new data were created in this review.

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
