# Peer review of "Electroceuticals and Magnetoceuticals in Gastroenterology"

_biomolecules, 2024, doi:10.3390/biom14070760_

Round 1

Reviewer 1 Report

Comments and Suggestions for Authors

This is a good review of an emerging field to treat GI disorders with electricity, magnetism, and acupuncture. This should be of interest to a wide range of readers.

Author Response

Thank you very much for your positive feedback regarding our paper.

Reviewer 2 Report

Comments and Suggestions for Authors

The authors address a very interesting and novel topic in gastroenterology which is the use of electroceuticals and magnetoceuticals. This type of treatment represent an attractive method particularly for GIT motility disorders and inflammatory conditions. The manuscript is well-written and organized. One comment to be addressed, is the need to write paragraphs on the disadvantages of these techniques, the safety and the drawbacks of their use.

Author Response

Comments and Suggestions for Authors:  The authors address a very interesting and novel topic in gastroenterology which is the use of electroceuticals and magnetoceuticals. This type of treatment represent an attractive method particularly for GIT motility disorders and inflammatory conditions. The manuscript is well-written and organized. One comment to be addressed, is the need to write paragraphs on the disadvantages of these techniques, the safety and the drawbacks of their use.

Response: Thank you for your positive feedback and constructive comment. In response to your suggestion, we have added a comprehensive section on the disadvantages, including stimulation parameters, safety concerns, and drawbacks of electroceuticals and magnetoceuticals. The added part is as follows:

Disadvantages and Drawbacks of Electroceuticals and Magnetoceuticals

Although electroceuticals and magnetoceuticals offer promising treatment options for GI disorders, several limitations and potential drawbacks should be considered. One significant limitation is that most of the clinical studies on vagal nerve electrical and magnetic neuromodulation involve small sample sizes and are often conducted as single-center studies. This can limit the generalizability and robustness of the findings. Another important limitation is the variability in patient response to these treatments. Individual differences in anatomy, disease severity, and concurrent medications can influence the efficacy of these neuromodulation techniques, leading to inconsistent outcomes among patients. Additionally, the precision required for device placement and parameter settings can pose challenges, as slight deviations may result in suboptimal therapeutic effects or adverse outcomes.

Stimulation Parameters

Stimulation parameters, including frequency, intensity, and duration, are critical for the success of neuromodulation techniques. Although there were recommended parameters in small trials from each single center, there is currently no consensus on the optimal parameters, necessitating customization for each patient. This variability can significantly impact treatment outcomes, complicating efforts to standardize neuromodulation therapies.

Safety Concerns

Although the safety profile of electroceuticals and magnetoceuticals is generally favorable, certain risks and adverse events may be associated with their use. For electroceuticals, potential side effects include skin irritation or burns at the electrode sites, discomfort during stimulation, and, in rare instances, interference with implanted medical devices such as pacemakers. Magnetic neuromodulation techniques may cause transient headaches, dizziness, or scalp discomfort. Additionally, there is a theoretical risk of inducing seizures, particularly in patients with a history of epilepsy or other neurological conditions.

Drawbacks

Moreover, the requirement for specialized equipment and trained personnel to administer these therapies can limit accessibility and increase the cost of treatment.

Reviewer 3 Report

Comments and Suggestions for Authors

It is an interesting and well written paper on a current topic: neuromodulation in Gastroenterology. The field is well covered. I have only minor comments.

- The description of the mechanisms of action of VNM is rather synthetic and needs more development.

- The authors need to further discuss stimulation parameters that are a potential limitation in the effectiveness of neuromodulation.

Author Response

Comments and Suggestions for Authors: It is an interesting and well written paper on a current topic: neuromodulation in Gastroenterology. The field is well covered. I have only minor comments.

- The description of the mechanisms of action of VNM is rather synthetic and needs more development.

  • The authors need to further discuss stimulation parameters that are a potential limitation in the effectiveness of neuromodulation.

Response: Thank you for your good comments. In response to your suggestions, we have made the following modifications:

1. We have revised the part of the mechanisms underlying VNM to be clearer and more detailed.

2. We have included a discussion of disadvantages, including stimulation parameters, in Section 5, "Disadvantages and Drawbacks of Electroceuticals and Magnetoceuticals," as follows:

Disadvantages and Drawbacks of Electroceuticals and Magnetoceuticals

Although electroceuticals and magnetoceuticals offer promising treatment options for GI disorders, several limitations and potential drawbacks should be considered. One significant limitation is that most of the clinical studies on vagal nerve electrical and magnetic neuromodulation involve small sample sizes and are often conducted as single-center studies. This can limit the generalizability and robustness of the findings. Another important limitation is the variability in patient response to these treatments. Individual differences in anatomy, disease severity, and concurrent medications can influence the efficacy of these neuromodulation techniques, leading to inconsistent outcomes among patients. Additionally, the precision required for device placement and parameter settings can pose challenges, as slight deviations may result in suboptimal therapeutic effects or adverse outcomes.

Stimulation Parameters

Stimulation parameters, including frequency, intensity, and duration, are critical for the success of neuromodulation techniques. Although there were recommended parameters in small trials from each single center, there is currently no consensus on the optimal parameters, necessitating customization for each patient. This variability can significantly impact treatment outcomes, complicating efforts to standardize neuromodulation therapies.

Safety Concerns

Although the safety profile of electroceuticals and magnetoceuticals is generally favorable, certain risks and adverse events may be associated with their use. For electroceuticals, potential side effects include skin irritation or burns at the electrode sites, discomfort during stimulation, and, in rare instances, interference with implanted medical devices such as pacemakers. Magnetic neuromodulation techniques may cause transient headaches, dizziness, or scalp discomfort. Additionally, there is a theoretical risk of inducing seizures, particularly in patients with a history of epilepsy or other neurological conditions.

 Drawbacks

Moreover, the requirement for specialized equipment and trained personnel to administer these therapies can limit accessibility and increase the cost of treatment.